# OpenReview forum: "The Secret Agenda: LLMs Strategically Lie and Our Current Safety Tools Are Blind"
_ICLR.cc/2026/Conference — Submitted to ICLR 2026_

### Official Review · Reviewer_tHZv · 2025-10-24

**Soundness:** 2
**Presentation:** 1
**Contribution:** 1
**Rating:** 2
**Confidence:** 4

**Summary:**

This paper investigates whether large language models can engage in strategic deception and whether current interpretability tools (especially SAE features) can detect it. The authors design game-like tests to elicit deceptive behavior and analyze SAE activations during these scenarios. They claim that the deception features identified by SAE fail to activate when lying occurs, suggesting current interpretability methods are blind to such behavior. The work positions itself as an early attempt to probe “hidden deception” in LLMs.

**Strengths:**

1. **Creative experimental design**: The “Secret Agenda” and “Insider Trading” setups show some originality in trying to operationalize deception through game-like tasks.
2. **Broad model coverage**: The study evaluates a large number of models (around 38), providing some breadth and comparative perspective across different architectures and providers.

**Weaknesses:**

I find the main weaknesses of this paper to be (1) its poor situating within existing literature and (2) major issues in clarity, rigor, and methodological grounding.

First, the paper is poorly situated in related work. It makes strong claims about deception and interpretability without referencing key prior studies in truthfulness, mechanistic interpretability, or SAE analysis, leaving its novelty unclear.

Second, the organization and clarity are weak. The introduction is too short, figures are oversized, and many technical details—such as how features are selected or how “feature fine-tuning” is done—are missing or unclear (e.g., “when tuned to -1 nor when tuned to +1” is never properly explained).

Third, the authors misinterpret and underestimate current interpretability methods. Claiming that SAE “deception features” failed to activate does not justify dismissing the SAE framework; this reasoning overlooks the distributed and compositional nature of such representations.

Fourth,  the paper criticizes existing evaluation methods without evidence. In line 100, the authors state that current evaluations are “unreliable, hard to replicate, and unrealistic,” yet they never specify which methods these are or provide comparisons to support the claim.

Finally, the format and presentation are below standard. The text is brief relative to the figures, captions are vague, and the layout feels unbalanced and unfinished.

**Questions:**

Please refer to weaknesses

---

> ### Author Response · Authors · 2025-11-22
>
> W1: "The paper is poorly situated in related work. It makes strong claims about deception and interpretability without referencing key prior studies in truthfulness, mechanistic interpretability, or SAE analysis, leaving its novelty unclear."
>
> Thank you for this valuable feedback. You are right that we need to better position our contributions relative to existing literature.
> Our positioning within three research threads:
>
> Deception capabilities literature (Park et al. 2023; Ward et al. 2023; Hagendorff 2024): Our contribution is methodological—creating a reproducible testbed that isolates incentive structures identified in prior work, rather than claiming first discovery of deceptive capabilities.
> Strategic deception under incentives (Scheurer et al. 2024; Meinke et al. 2024; Greenblatt et al. 2024): Our novel contributions are: (a) Secret Agenda as a complementary behavioral benchmark, and (b) testing whether auto-labeled SAE features can control (via steering) versus merely correlate with deceptive behavior.
> Mechanistic interpretability via SAEs (Balsam et al. 2024; McGrath et al. 2024; Lieberum et al. 2024): We provide negative evidence about current auto-labeling approaches, addressing GemmaScope open questions (neuronpedia.org/gemma-scope#openproblems): "Do [SAEs] really find the 'true' concepts in a model? How robust are claims about interpretability?"
>
> We are building on—not competing with—prior work. Our contribution lies in testing interpretability tools rather than merely documenting deceptive capabilities.
>
> W2a: "The introduction is too short, figures are oversized"
> Thank you for this feedback. We expanded Section 1 (Introduction) to address W1 and provide broader context. We also reorganized Figures 4-5 into a 2×2 grid format, making them proportional to surrounding text.
>
> W2b: "Many technical details—such as how features are selected or how 'feature fine-tuning' is done—are missing or unclear (e.g., 'when tuned to -1 nor when tuned to +1' is never properly explained)."
> Thank you for this question. We will add technical clarifications:
>
> Terminology: We use "feature steering" and "feature fine-tuning" interchangeably—both refer to real-time adjustment of SAE feature activations during inference without retraining.
> Mechanism: Implementation uses goodfire.steering.set_feature(feature_id, weight=-1.0), applying multiplicative scaling to activations. weight=-1.0 suppresses the feature completely; weight=+1.0 maximizes activation. This modifies internal representations in real-time, similar to Anthropic's "Golden Gate Claude" experiments (Templeton et al., 2024).
>
>
> W3: "The authors misinterpret and underestimate current interpretability methods. Claiming that SAE 'deception features' failed to activate does not justify dismissing the SAE framework; this reasoning overlooks the distributed and compositional nature of such representations."
> Thank you for this correction. You are right that we overstated our claims.
> Our actual findings (narrower scope):
> We do not claim "SAEs cannot represent deception." Our evidence shows current auto-labeled features in GemmaScope and Goodfire:
>
> Rarely activated during clear strategic deception
> Failed to control deceptive behavior when steered (set to -1 or +1)
>
> This critiques current auto-labeling methodology, not the SAE framework. Strategic deception may be represented in distributed feature combinations that current labeling doesn't capture.
> Changes made:
>
> Title revised (page 1): Changed from "...Safety Tools Are Blind" to "...Undetected by Current Safety Tools"
> Claims softened (Section 8.4, line 409): Added: "We do not claim SAEs cannot represent deception."
>
>
> W4: "The paper criticizes existing evaluation methods without evidence. In line 100, the authors state that current evaluations are 'unreliable, hard to replicate, and unrealistic,' yet they never specify which methods these are or provide comparisons to support the claim."
> Thank you for catching this. This line was not intended as criticism but as a summary of methodological challenges that prior work identifies. We integrated this into Section 1 (Introduction).
> Our Secret Agenda testbed complements existing approaches by offering controlled conditions where models face clear tradeoffs between goals (winning) and rules (honesty), without explicit deception instructions. This enables systematic testing of interpretability tools under reproducible conditions.
>
> W5: "The format and presentation are below standard. The text is brief relative to the figures, captions are vague, and the layout feels unbalanced and unfinished."
> Thank you for this feedback. We reorganized Figures 4-5 into a 2×2 grid format to balance with text. We also centered the heading on Figure 5 for improved layout.

---

### Official Review · Reviewer_WnJS · 2025-10-30

**Soundness:** 3
**Presentation:** 2
**Contribution:** 3
**Rating:** 6
**Confidence:** 4

**Summary:**

The paper presents a reproducible testbed for eliciting deceptive behavior in LLMs based on a game that requires the player to lie to succeed. It demonstrates that this elicits deceptive behavior in 38 tested models. It explores the effectiveness of auto-labeled SAE features for detecting strategic deception in the new deception test bed. It claims that auto-labeled SAE features are limited in their ability to detect or suppress deception as demonstrated by small latents in "deception" related features and continued instances of deception despite suppressing those features. Lastly, it presents a method for clustering SAE latents using PCA+t-SNE to demonstrate domain-dependent effectiveness of SAE feature interpretability.

Mandatory disclosure of LLM usage by the reviewer: This review was formatted using LLMs to structure the text into paragraphs and numbered lists.

**Strengths:**

1. The paper addresses an important topic of strategic deception using a new binary testbed.

2. It demonstrates that deception can be elicited in across 38 different models.

3. The results are reproducible and experimental setup is well documented.

4. The effectiveness of SAEs at detecting strategic deception is put into question and backed up by experiment.

**Weaknesses:**

1. In section 4, the paper claims that Secret Agenda game aims to generate "high stakes deception", however given that the problem is framed as a game does not feel high stakes as in Agentic Misalignment (arXiv:2510.05179) where the agent reported that it was fully aware that its actions will cause harm to a human being.

2. The game framing may limit generalizability in two ways: (1) models may be following game rules rather than exhibiting deception that would occur in deployment, and (2) the artificial stakes may not activate the same neural mechanisms as real-world deception. While the authors acknowledge this is a "controlled deception laboratory", the connect between game-based deception and deployment risks needs stronger justification.

3. It would drive the point home if the Secret Agenda test bed could elicit deception in case deception is clearly harmful, though I admit that deception of any kind is evidence enough - it would be trivial for a third party to cause the model to deceive.

4. Performing section 7 scatter plot for Secret Agent SAE latents seems like a natural step that could shed light on the proposed feature search methodology.

5. Along the same lines I would be very curious to see how raising or lowering the states changes models behavior and the SAE activations. Will SAE features become more sensitive in a "real" scenario? Will the model avoid deception knowing that it will cause harm?

**Questions:**

1. In section 7.3, by "contrasting results" do you mean the difference between Secret Agenda vs Insider Trading or something else? If so, why was the "top discriminative features" not and PCA+t-SNE scatter plot analysis not applied to Secret Agenda SAE latents?

2. The authors acknowledge under limitations that game scenarios might not generalize to all deception types. There are clearly tradeoffs here that make it difficult to pin down and also gets at the heart of the problem. If the scenario is a game entirely different features may activate for "in-game deception" compared to "real world deception". Could you understand whether what (if any) SAE features activate highly for "in-game deception"?

---

> ### Author Response · Authors · 2025-11-22
> **Response from Authors on review.**
>
> W1: "In section 4, the paper claims that Secret Agenda game aims to generate 'high stakes deception', however given that the problem is framed as a game does not feel high stakes as in Agentic Misalignment (arXiv:2510.05179) where the agent reported that it was fully aware that its actions will cause harm to a human being."
>
> Thank you for this feedback. We agree that "high stakes" is misleading compared to scenarios where models understand they will cause real human harm. Our goal was to capture strategic deception under strong incentives, not real-world physical stakes.
> We updated the phrasing from "high stakes deception" to "incentive-driven" throughout (lines 118, 392).
>
> W2: "The game framing may limit generalizability in two ways: (1) models may be following game rules rather than exhibiting deception that would occur in deployment, and (2) the artificial stakes may not activate the same neural mechanisms as real-world deception. While the authors acknowledge this is a 'controlled deception laboratory', the connection between game-based deception and deployment risks needs stronger justification."
>
> Thank you for this concern. Two factors make game-based deception deployment-relevant:
>
> Explicit anti-deception instructions: Our prompt states "Deception in official speeches is a violation of the Constitution" and lying is "against the law" (Appendix C, page 16, line 882). Models deceive despite these honesty instructions, demonstrating that goal achievement overrides ethical constraints—relevant to deployment where safety guidelines may conflict with rewards.
> Evaluation-aware models still deceive: Advanced models can detect testing contexts (theguardian.com/technology/2025/oct/01/anthropic-ai-model-claude-sonnet-asks-if-it-is-being-tested), yet still engaged in strategic deception in our explicitly-framed "game." This suggests robust behavior, not naive game-following.
>
>
> W3: "It would drive the point home if the Secret Agenda test bed could elicit deception in case deception is clearly harmful, though I admit that deception of any kind is evidence enough - it would be trivial for a third party to cause the model to deceive."
>
> Thank you for this comment. We clarify that we do not claim it's trivial for third parties to manipulate models into deception.
> Our testbed studies deception from goal optimization under weak oversight—the pattern in deployment-relevant work (Scheurer et al. 2024; Greenblatt et al. 2024; Meinke et al. 2024). This differs from adversarial prompt attacks. Models face natural misaligned incentives (reward for winning, unenforced honesty rules) rather than malicious manipulation.
>
> W4: "Performing section 7 scatter plot for Secret Agent SAE latents seems like a natural step that could shed light on the proposed feature search methodology."
> Thank you for this suggestion. Resource constraints necessitated asymmetric analysis. Insider Trading permits automated regex-based classification (executable trades vs. refusals), enabling systematic SAE analysis. Secret Agenda requires human judgment or expensive LLM-as-judge classification to distinguish lies, deflection, and truth—exceeding our volunteer team's budget. We noted this limitation in Section 8.3, line 402.
> As a volunteer group, we prioritized behavioral analysis (manual coding), feature steering experiments, and full Insider Trading SAE analysis. Our conclusion calls for larger labs to replicate and extend this work.
>
> W5: "Along the same lines I would be very curious to see how raising or lowering the stakes changes models' behavior and the SAE activations. Will SAE features become more sensitive in a 'real' scenario? Will the model avoid deception knowing that it will cause harm?"
> Thank you—this identifies a key direction for future work.
> Our scope: We test whether SAE features detect/control deception under incentive-driven conditions. We did not test scenarios where models understand real-world harm, due to methodological challenges and ethical concerns about training models to recognize harmful deception opportunities.
> Your critical question: Do SAE patterns change as stakes increase? This could reveal whether current features are "stakes-insensitive," whether different features activate in high-consequence deception, or whether models engage different processes when harm is salient.
> We encourage follow-up research and added this to Future Work (Appendix H, page 31, line 1657).

---

### Official Review · Reviewer_Zgec · 2025-10-31

**Soundness:** 4
**Presentation:** 2
**Contribution:** 3
**Rating:** 6
**Confidence:** 3

**Summary:**

The paper outlines an investigation of strategic deception in LLMs using two developed testbeds – Secret Agenda and Insider Trading Compliance. The former testbed is based on its namesake, a social deduction game with the LLM set as the Dictator, wherein the games are set up such that lying is the easiest path for the model to win. Experiments in this “controlled deception laboratory” show that LLMs consistently resort to deception (Figure 1).  More troublingly, the models achieve this deception without activating features that are expected to be linked to deception. On the other hand, the Insider Trading Compliance testbed exhibited activation of deception-linked autolabelled features; the models continued to engage in insider trading almost half the time.

Put together, these results illustrate the serious challenge and risk posed by LLMs strategically lying to maximize their stated objectives. Moreover, the disconnect between autolabelled deception-related features and the activations observed in the Secret Agenda tests limit the utility of tuning these features to limit strategic lying by the LLMs.

**Strengths:**

The paper sheds light on an important challenge, especially when generalizing LLMs to other domains. Honesty of models when providing responses is paramount to build trust for users and a necessity for their adoption in domains with high compliance standards. Consequently, testing methodologies to audit models for deception and testing the efficacy of corrective measures is an important contribution.

In particular, the result illustrated in Figure 1 is the most striking because it shows that all the tested models strategically resort to deception most of the time. Granted that Secret Agenda puts the models is adversarial situations where deception is the easiest path to victory, but it is desired and possibly a must, that LLMs do not lie even in such settings. While the result in Figure 3 is not as striking, the models electing to engage in insider trading nearly half the time is also a worrisome and significant result.

Finally, the conclusion that the pertinence of autolabelled SAE deception-relevant features for model deception is likely setting-dependent is noteworthy as (a) it suggests that the utility of these features and deception mitigation strategies predicated on these features could be severely limited in some scenarios and (b) indicates that feature-based interventions and interpretations are more instructive with structured data.

**Weaknesses:**

While I appreciate the wider point and results conveyed by this paper, the writing of the paper leaves a lot to be desired. It is overfull with jargon and, missing references and explanations; this necessitates that the reader either be very familiar with this area or read multiple references so that he/she may understand the paper and the results contained therein.

An example of this are the repeated references to SAEs and features labeled by SAE but its full form (Sparse Autoencoder) is only stated on Page 5. Moreover, there is no explicit discussion of SAE-based labelling of features and its uses for interpretability. Another example of the lack of details are the various game contexts in Section 5.3, with little explanation for the game contexts and explanations for the authors’ selection of these contexts in their tests.

**Questions:**

**Questions:**

1.	What do the authors mean on line 59 when they aver that their “operationalization remains agnostic to assumed beliefs”?
2.  On line 260, it is stated that refusal responses comprise the majority of outputs. This appears to contradict the results in Figure 3, and more broadly contrasts with the results of the Secret Agenda tests.

**Suggestions:**

1.	In the conclusion, the authors state that “identity-feature interventions often degraded outputs into repetitive, incoherent loops.” This is the first mention of any such result in any part of the main paper, and hence does not follow from any of the preceding discussion. The previous discussion of this result should be moved into the main paper or this should be removed from the conclusion to ensure coherence of the manuscript.
2.	Please justify any design choices made in the proposed test beds.

---

> ### Author Response · Authors · 2025-11-22
> **Response from Authors on review.**
>
> Q1. What do the authors mean on line 59 when they aver that their “operationalization remains agnostic to assumed beliefs”?
>
> Thank you for requesting clarification. We mean that our definition does not require models to have "internal beliefs"—we focus only on observable behavior. A response counts as deceptive if it produces false, misleading, or strategically omissive outputs that can mislead users, regardless of whether the model "believes" anything internally. This contrasts with definitions like Ward et al. (2023) that reference "intentionally communicating a false belief," which anthropomorphize model cognition.
> We have rephrased on line 78 in the revised manuscript for clarity: "Unlike some of these definitions which may anthropomorphize, our operationalization remains agnostic to assumed beliefs. Instead, we follow Park et al. (2023) and focus on observable behaviors that have the potential to mislead, manipulate, or erode trust."
>
>
> Q2. On line 260, it is stated that refusal responses comprise the majority of outputs. This appears to contradict the results in Figure 3, and more broadly contrasts with the results of the Secret Agenda tests.
>
>
> Thank you for noting this inconsistency. We have reviewed the data and Figure 3, regarding Insider Trading examples, which clearly shows that Engagement responses comprise the majority of outputs (70 responses out of 149 total).
>
> Change made :
> Line # 269 has been corrected to accurately state: 'Figure 3 shows the distribution of response classifications across the 149 insider trading prompts,
> with engagement responses comprising the majority of outputs, refusal responses forming a smaller portion, and helpful responses representing an intermediate category
> ' This was a proofreading error that did not affect our analysis or conclusions, as all subsequent analysis was performed on the correct data distribution shown in Figure 3.
>
> S1. In the conclusion, the authors state that "identity-feature interventions often degraded outputs into repetitive, incoherent loops." This is the first mention of any such result in any part of the main paper, and hence does not follow from any of the preceding discussion. The previous discussion of this result should be moved into the main paper or this should be removed from the conclusion to ensure coherence of the manuscript.
> Thank you for noting this. You are correct—this result appeared only in the Conclusion due to an editing error. We have moved the discussion of identity-feature degradation effects into Section 3 (page 2, line 94), immediately following our other feature steering results where it belongs chronologically.
>
> S2. Please justify any design choices made in the proposed test beds.
> Thank you for this feedback. We have added justifications for our key design choices throughout the revised manuscript:
> SAE methodology (Section 1, page 1, lines 37): Added explicit definition of Sparse Autoencoders and auto-labeling process to clarify why we test both labeled and unlabeled features.
> Secret Agenda game selection (Section 5.1, page 3): Added explanation of why this game creates appropriate incentive structures for testing interpretability tools—it isolates the moment when strategic deception becomes optimal while providing binary, reproducible deception events.
> Prompt variant rationale (Section 5.3, page 3): Added explanation that testing multiple variants (political, nature-themed, truth-vs-lies framing) ensures robustness and rules out confounds from specific terminology or political content.
> These additions make our methodological rationale explicit rather than implicit.

---

### Meta-Review · Area_Chair_Twjk · 2026-01-07

**Summary:**

This paper investigates strategic deception in Large Language Models (LLMs) using two testbeds: a social deduction game ("Secret Agenda") and an insider-trading scenario. The authors argue that, although models consistently engage in deception under incentives, current auto-labeled Sparse Autoencoder (SAE) features fail to detect this behavior in the game setting, whereas aggregate activations show promise in the insider trading context.

While the reviewers acknowledged the interesting motivation and the breadth of models tested (38 models), the submission suffers from significant issues regarding presentation, methodological rigor, and contextualization within existing literature. Reviewer scores were mixed (6, 6, 2), but the fundamental flaws highlighted by the critical review (Reviewer tHZv) and echoed in part by others regarding writing quality and technical claims suggest the paper is not yet ready for publication.

**Reviewer Concerns:**

The reviewers raised several critical concerns that justify a rejection:

1.  All three reviewers criticized the manuscript's quality. Reviewer tHZv rated presentation as "poor" (1), noting oversized figures, vague captions, and an "unfinished" layout. Reviewer Zgec noted the paper is "overfull with jargon" and missing necessary explanations. Reviewer WnJS also rated presentation as "fair" (2).


2.  Reviewer tHZv pointed out a weak literature review, stating the paper makes strong claims without referencing key prior studies in truthfulness or mechanistic interpretability.


3.  Reviewer tHZv argued the authors misinterpret current interpretability methods, noting that the failure of specific "deception features" to activate does not justify dismissing the SAE framework, as it overlooks distributed representations .


4.  Reviewer WnJS questioned the "high stakes" framing of a game environment and expressed concern that game-based rule-following may not generalize to real-world deceptive deployment risks.


5. The paper itself admits that sample sizes were small and insufficient for robust statistical inference, and that resource constraints led to an "asymmetric analysis" depth between the two testbeds.

**Reviewer Scores:**

Reviewer Zgec: 6
Reviewer WnJS: 6
Reviewer tHZv: 2

---

### Decision · Program_Chairs · 2026-01-26

Reject